# Laboratory Evaluation of Low-Cost Optical Particle Counters for Environmental and Occupational Exposures

**DOI:** 10.3390/s21124146

**Published:** 2021-06-17

**Authors:** Sinan Sousan, Swastika Regmi, Yoo Min Park

**Affiliations:** 1Department of Public Health, Brody School of Medicine, East Carolina University, Greenville, NC 27834, USA; 2North Carolina Agromedicine Institute, Greenville, NC 27834, USA; 3Environmental Health Sciences Program, Department of Health Education and Promotion, College of Health and Human Performance, East Carolina University, Greenville, NC 27834, USA; swastika.bph@gmail.com; 4Department of Geography, Planning, and Environment, East Carolina University, Greenville, NC 27858, USA; parky19@ecu.edu

**Keywords:** particulate matter, OPC, OPC-N3, SPS30, AirBeam2, PMS A003, sensor calibration, environmental monitoring, occupational monitoring, low-cost sensors

## Abstract

Low-cost optical particle counters effectively measure particulate matter (PM) mass concentrations once calibrated. Sensor calibration can be established by deriving a linear regression model by performing side-by-side measurements with a reference instrument. However, calibration differences between environmental and occupational settings have not been demonstrated. This study evaluated four commercially available, low-cost PM sensors (OPC-N3, SPS30, AirBeam2, and PMS A003) in both settings. The mass concentrations of three aerosols (salt, Arizona road dust, and Poly-alpha-olefin-4 oil) were measured and compared with a reference instrument. OPC-N3 and SPS30 were highly correlated (*r* = 0.99) with the reference instrument for all aerosol types in environmental settings. In occupational settings, SPS30, AirBeam2, and PMS A003 exhibited high correlation (>0.96), but the OPC-N3 correlation varied (*r* = 0.88–1.00). Response significantly (*p* < 0.001) varied between environmental and occupational settings for most particle sizes and aerosol types. Biases varied by particle size and aerosol type. SPS30 and OPC-N3 exhibited low bias for environmental settings, but all of the sensors showed a high bias for occupational settings. For intra-instrumental precision, SPS30 exhibited high precision for salt for both settings compared to the other low-cost sensors and aerosol types. These findings suggest that SPS30 and OPC-N3 can provide a reasonable estimate of PM mass concentrations if calibrated differently for environmental and occupational settings using site-specific calibration factors.

## 1. Introduction

Air pollution is one of the leading causes of death in both developing and economically developed countries. The World Health Organization [1] estimated that 7 million deaths worldwide annually are due to air pollution, which results in stroke, ischemic heart disease, lung cancer, chronic obstructive pulmonary disease (COPD), and acute respiratory infections. Particulate matter (PM) is one of the six criteria for air pollutants regulated under the National Ambient Air Quality Standards established by the U.S. Environmental Protection Agency (EPA). PM consists of liquid and solid particles suspended in the air we breathe, such as oil, dust, sea salt, forest fire ash, black carbon, and sulfates [2]. PM poses a significant threat to human and environmental health. Exposure to PM is associated with various cardiovascular and respiratory diseases, and the overall impact of PM on the human body depends on particle size, concentration, and chemical composition [3]. Cardiovascular diseases associated with PM_2.5_ (particles 2.5 µm and smaller) include myocardial infarction, arrhythmia, stroke, and heart failure [4]. Brook et al. [4] showed that exposure to PM elevates cardiovascular disease risk, resulting in a higher morbidity and mortality rate. As lungs are the primary site of deposition of the inhaled PM [5], many epidemiological studies have reported the correlation between indoor and outdoor PM exposure and respiratory diseases such as COPD, lung cancer, pneumonia, and asthma [6,7]. Indoor exposure to a higher concentration of PM_2.5_ has been found to trigger bronchial and asthmatic symptoms [6]. Similarly, exposure to coal dust particulates (coarse particles), which is an occupational hazard, is associated with severe lung diseases such as coal workers’ pneumoconiosis, silicosis, and COPD [8].

In the United States, the EPA regulates the concentration of various air pollutants, including PM. PM_1_, PM_2.5_, and PM_10_ are the major particle sizes, and they have an aerodynamic diameter of 1 µm and smaller, 2.5 µm and smaller, and 10 µm and smaller, respectively. High concentrations of PM_2.5_ are of the greatest concern because the fine particle size can pass through the upper respiratory tract and reach deep into the lungs, and particles less than 0.1 µm can translocate into the bloodstream [9]. EPA [10] limits for PM_2.5_ are 15 μg/m^3^ (annually) and 35 μg/m^3^ (daily), compared to 150 μg/m^3^ (daily) for PM_10_. Similarly, the Occupational Safety and Health Administration (OSHA) sets standards for PM and other occupational hazards to guarantee a safe and healthy working environment. The 8 h time-weighted standard for respirable dust particulates (particles 4 µm and smaller) that are otherwise not regulated is 5 mg/m^3^ (or 5000 μg/m^3^) [11].

It is essential to understand the differences in both sources and short- and long-term effects between environmental and occupational settings. Environmental conditions are susceptible to a mixture of different pollutants that coexist [12]. PM could be generated directly from the source or created due to a series of atmospheric chemical reactions. In addition, environmental pollutants from power source generators, factories, and automobiles are highly diffusive in the atmosphere, and countries like the United States are monitored to ensure low concentrations from the source by establishing environmental regulations. Due to the diffusive nature of environmental pollutants, PM transportation is known to occur locally or across continents. In contrast, occupational pollutants could be generated through outdoor activities such as construction work [13] or landscaping [14], or indoor activities such as welding [11] or mining [15]. These sources are associated with high PM concentrations within short periods that affect on-site workers and put them at risk. Occupational concentrations are a hundred or thousand times higher than environmental concentrations, especially at the source [11]. Therefore, on-site control methods are established to mitigate possible immediate effects, such as ventilation systems and personal protective equipment.

To meet the Federal Clean Air Act’s National Ambient Air Quality Standards, federal reference methods (FRMs) and federal equivalent methods are used in various air monitoring stations throughout the United States to monitor six criteria air pollutants, including PM [16]. FRM sites consist of filter-based gravimetric methods that provide precise and accurate measurements for PM concentrations [17]. FRM is the gold standard method for air quality monitoring, and it uses filter samples collected over 24 h, which are then weighed and analyzed in a laboratory. However, this method does not provide real-time information. Moreover, due to their large size and high operating costs [17], these devices are used in sparsely located air monitoring stations spread throughout the United States. Therefore, these devices do not provide adequate spatial and temporal information.

Recently, there has been an increase in the use of low-cost PM sensors to monitor health hazards, especially in developed countries [18]. These devices are easy to operate, portable, compact in size, and relatively inexpensive. Optical particle counters (OPCs) are the most widely used devices for counting and sizing particles ranging from 0.25 µm to several microns. OPCs measure the amount of light scattered by individual particles as they navigate through a beam of light. A fraction of the scattered light is collected and directed to a photodetector, where it is converted to a voltage pulse [19]. The particle size is then determined from the magnitude of this voltage pulse using a calibration curve. Several studies have been conducted to evaluate the performance of low-cost sensors in laboratory and field settings; however, the results have been somewhat mixed depending on the type of sensor used [3,11,20,21,22,23].

Sousan et al. [24] evaluated the OPC-N2 by Alphasense using three aerosol types: salt, welding fume, and Arizona road dust (ARD). They found high linearity (r = 0.97) between the mass concentrations measured by OPC-N2 and reference instruments (SMPS and APS sensors) for all aerosol types. The authors also found moderate to high precision among OPC-N2 sensors used in the experiment (CV = 4.4% to 16%). Tryner et al. [25] evaluated SPS30 sensors for various aerosol types, such as ammonium sulfate, ARD, polystyrene latex spheres (PSL), and wood smoke, in a laboratory setting. The authors compared the results with those of the TEOM, SMPS, and APS reference sensors and reported that the PM_2.5_ mass concentration calculated by SPS30 was highly linear with the results obtained from the reference instruments for all aerosol types at a concentration greater than 540 µg/m^3^. Moreover, precision among SPS30 sensors was high for all aerosol types. Levy Zamora et al. [3] evaluated the PMS A003 sensor using various aerosol sources (incense, oleic acid, sodium chloride (NaCl), talcum powder, PSL particles, etc.) and found a high correlation (R^2^ > 0.86) compared to the reference instrument for PM_2.5_ mass concentration for all aerosol types. The authors also observed a high degree of precision among PMS A003 sensors measuring PM_2.5_ mass concentrations.

However, no studies have yet evaluated the accuracy of sensors for both environmental and occupational settings at the same time. In addition, no studies have yet provided information on whether specific sensors should be calibrated differently for environmental and occupational settings. This study evaluates the performance of low-cost sensors in laboratory settings to compare the difference in performance between environmental and occupational exposures. The study’s specific objectives are to (1) compare the performance of low-cost OPCs for salt, ARD, and Poly-alpha-olefin-4 (PAO-4) oil with the reference instruments; (2) provide recommendations for the selection of low-cost sensors suitable for environmental and occupational exposure; and (3) determine if the sensors should be calibrated differently for environmental and occupational exposure for the same aerosol type.

## 2. Materials and Methods

### 2.1. Aerosol Sensors

The low-cost OPCs evaluated in this study were OPC-N3 (Alphasense, Essex, UK), SPS30 (Sensirion, Stäfa, Switzerland), AirBeam2 (HabitatMap, Brooklyn, NY, USA), and PMS A003 (Plantower, Beijing, China). All the low-cost sensors were new. They were chosen based on their low cost (<$500) compared to the reference instrument (>$30,000). All these PM sensors follow the same light scattering principle to measure PM_1_, PM_2.5_, and PM_10_ mass concentrations in real time; however, they differed in certain specifications, as shown in Table 1.

#### 2.1.1. OPC-N3

The OPC-N3 sensor uses a laser beam to count the particles, which are classified into 24 bins ranging from 0.35 µm to 40 µm in diameter [26]. Those particle counts are then converted into mass concentrations of PM_1_, PM_2.5_, PM_4_, and PM_10_ based on the embedded algorithms. OPC-N3 also measures temperature, relative humidity, and mass concentrations of particles up to 2000 µg/m^3^. OPC-N3 has an active sensor that uses an internal fan to sample the air. The device has no internal battery; therefore, it must be connected to an external power source for operation. It also has a built-in micro SD card for data storage.

#### 2.1.2. SPS30

SPS30 classifies particles into five different bins ranging from 0.3 µm to 10 µm in diameter. Those particle counts are then converted to mass concentrations (PM_1_, PM_2.5_, PM_4_, and PM_10_) and number concentration (PM_0.5_, PM_1_, PM_2.5_, PM_4_, and PM_10_) using embedded advanced proprietary algorithms [27]. Unlike other low-cost PM sensors, SPS30 is developed with proprietary contamination-resistant technology and advanced binning technologies that provide more precise measurements for a lifetime of over 10 years. The device performs self-cleaning, and no maintenance is required. SPS30 has an internal fan for sampling purposes and can measure the concentration of particles ranging from 1 to 1000 µg/m^3^. The device has no internal storage, and is operated using a computer and software provided by the manufacturer. It must be connected to an external power source for operation due to the lack of an internal battery. Additionally, it does not report temperature or relative humidity.

#### 2.1.3. AirBeam2

AirBeam2 uses the PMS 7003 sensor, which is from the Plantower family that classifies particles into six different bins ranging from 0.3 µm to 10 µm in diameter [28]. To report the mass concentrations of PM_1_, PM_2.5_, and PM_10_, AirBeam2 uses the calibration equations developed by fitting the PurpleAir-I and PurpleAir-II measurements to the GRIMM EDM180, which was further confirmed by comparing the AirBeam2 data with a TSI DustTrak DRX Aerosol Monitor 8533. The PurpleAir sensors also contain Plantower PM sensors. AirBeam2 also measures temperature, relative humidity, and the concentration of particles ranging from 0 to 1000 µg/m^3^. It has an internal battery that can power the device continuously for 10 h when fully charged. However, it does not have any internal storage. Therefore, mobile sessions are communicated to an AirCasting website via WiFi or cellular data. Once the session is completed, data can be retrieved via email from the AirCasting website for analysis. AirBeam2 is considered an IoT device that wirelessly transfers data to a network [29]. However, AirCasting provides an open-source code for the AirBeam2 so that users can change the calibration equation for the PMS 7003 if needed.

#### 2.1.4. PMS A003

The PMS A003 is also a member of the Plantower family and measures particles ranging from 0 to 1000 µg/m^3^. For sampling purposes, it has an internal fan that draws in the air and passes it through a laser beam. It measures the amount of light scattered by individual particles as they navigate through a beam of light [3]. The particles are then divided into six different bins ranging from 0.3 µm to 10.0 µm in diameter. Those particle counts are then converted to mass concentrations (PM_1_, PM_2.5_, and PM_10_) using the embedded algorithms. PMS A003 reports “standard” and “atmospheric” PM mass concentrations based on the particle density. The standard PM mass concentration is used in specific settings such as industrial settings, whereas the atmospheric PM mass concentration is used for ambient air settings. Because PMS A003 has no internal battery, it must be connected to an external power source for operation. The PMS A003 is a standalone sensor that requires a microcomputer for operation. Therefore, we used the GeoAir1 device that contains the PMS A003 to store and obtain the output of the PMS A003 sensor. Details on the GeoAir1 device can be found in Park’s [30] paper. The AirBeam2 also contains a Plantower PM sensor. However, the study compares the AirBeam2 that indirectly estimates mass concentration using a linear regression model with the PMS A003 that estimates mass concentrations by binning the particles and calculating the respective mass concentration for each size.

#### 2.1.5. pDR-1500

The pDR-1500 sensor is a medium-cost ($7000) photometer developed by Thermo Fisher Scientific Inc. (Waltham, MA, USA) to measure PM concentration (Table 1). pDR-1500 has different cyclones with specific cut-points, which can be used interchangeably to measure different particle sizes in different environments [31]. In addition, pDR-1500 has a built-in 37 mm filter holder that collects the particles for gravimetric analysis. pDR-1500 also reports temperature, relative humidity, and high concentrations of PM ranging from 0.001 to 400,000 µg/m^3^. It can be powered by an internal battery or by AC power. It has internal storage, and the data can be retrieved easily after completion of the experiment.

#### 2.1.6. Reference Instrument

The GRIMM Mini Wide-Range Aerosol Spectrometer (MiniWRAS 1371, Grimm Aerosol Technik, Ainring, Germany) is a high-cost (>$30,000) spectrometer that can measure particle size in different ranges between 0.01 μm and 35.00 μm [32]. The device divides the particles into 41 bins and multiple mass concentrations of PM_1_, PM_2.5_, and PM_10_. The MiniWRAS was chosen as a reference instrument in this study because it has almost twice the bins as the OPC-N3, and seven times the bin count for the other low-cost sensors. Increasing the number of bins helps to accurately estimate different particle sizes. In addition, the MiniWRAS laser operates at different powers of 40 mW for detecting particles smaller than 2.5 µm and 0.5 mW for larger particles. The GRIMM MiniWRAS uses a corona charger to measure electrical mobility for particles smaller than 0.25 μm, which is a size range that cannot be measured with optical science. It uses an optical sensor for particles larger than 0.25 μm. Overall, the MiniWRAS parts are high-cost and reliable, with built-in hardware and software that maintain the device.

### 2.2. Experimental Setup

#### 2.2.1. Chamber Description

Experiments were performed at the East Carolina University Aerosol laboratory inside a controlled exposure chamber with dimensions of 1.83 m × 0.66 m × 0.66 m (L × W × H). The chamber was built with airtight plexiglass. It consisted of a mixing zone and a sampling zone separated by a honeycomb straightening section (AS100, Rusken, Grandview, MO, USA). The 0.61 m × 0.61 m × 0.66 m (L × W × H) mixing zone contained an inlet at the bottom for aerosol generation. Two small fans were placed inside the mixing zone to mix the generated aerosol. Particle-free air was supplied to the mixing zone using two industrial-sized high-efficiency particulate air (HEPA) filters with 99.99% efficiency. The honeycomb sheet allowed for even distribution of laminar airflow with no dead zones inside the sampling zone. The chamber schematic and actual photo are shown in Figure 1a and Appendix A, respectively.

The 0.61 m × 0.61 m × 0.66 m (L × W × H) sampling zone consisted of an outlet for the exhaust air. The outlet was attached to a vacuum with two HEPA filters that pulled in the exhaust air and filtered it. A valve near the vacuum was used to adjust the flow rate and control the PM concentration inside the sampling zone. Prior to the experiment, the mixing performance of the chamber was tested by placing three SPS30 sensors at three different locations (left, center, and right corner of the sampling zone) to demonstrate its homogeneity. During the experiment, the low-cost OPCs and medium-cost aerosol photometer were directly placed inside the sampling zone as shown in Appendix A. The reference instrument was placed outside the chamber with an isokinetic sampling probe connected to the sampling zone. Inside the chamber, the temperature and relative humidity were kept relatively constant.

#### 2.2.2. Aerosol Generation

For the experiments, three different aerosols—Salt, ARD, and PAO-4 oil—were generated using three different aerosol generation systems (Figure 1b). The different generation methods were conducted under controlled temperature (23 ± 2 °C) and relative humidity conditions (35 ± 5%). For each aerosol type, two experiments were conducted to simulate environmental and occupational concentrations. Salt was chosen because it is the most widely used and safest aerosol source for evaluating OPC sensors [11]. Salt was generated using the Aeroneb Pro-X Controller (Aerogen, Chicago, IL, USA) using a 0.9% solution of NaCl, which was prepared in the laboratory (Figure 1b, Part I). A mass flow controller (MFC) was supplied with particle-free air using a five-stage desiccant system that controlled the airflow into the silica column. The salt was then passed through the silica column to remove the moisture, and the dry salt particles entered the mixing zone. The generated moisture-free aerosol was then mixed with particle-free air to achieve the desired concentration. The amount of salt generated was controlled by altering Aeroneb output and by changing the flow rate (2–3 L/min) using the MFC.

ARD (PTI ID: 13328B; Powder Technology Inc., Arden Hills, MN, USA) was chosen as the second aerosol as it is commonly used in the calibration of PM sensors and also represents the coarse mineral dust present in many environmental and occupational settings [11]. The dust was generated using a Vilnius aerosol generator (VAG; CH Technologies, Westwood, NJ, USA), as shown in Figure 1b, Part II. The VAG was a dry powder dispenser that produced aerosol concentrations from 1 to 2500 mg/m^3^. It consisted of three units: a controller, a dispenser, and a monitor. Per the manufacturer’s instruction, 1.33 g of dry dust was loaded into the dust chamber of the VAG all at once. Then, particle-free air was supplied at a flow rate of 6–13 L/min using a five-stage desiccant operated by an MFC connected to the VAG dispenser. The VAG dispenser outlet was attached to a metal tube connected to the mixing chamber, where ARD was generated and mixed with particle-free air.

PAO-4 oil was the final particulate selected for this study because it is a U.S. Food and Drug Administration-approved substitute for dioctyl phthalate oil, and the National Institute for Occupational Safety and Health (NIOSH) uses it to test R and P series filters to determine their efficacy [33,34]. It is also used for testing HEPA and ultralow-penetration air filters [35]. A TDA-4B Lite aerosol generator (Air Techniques International, Owings Mills, MD, USA) and PAO-4 oil (Air Techniques International) were used for the oil aerosolization (Figure 1b, Part III). Following the manufacturer’s instructions, two-thirds of the oil generator was filled with PAO-4 oil. Then, particle-free air was supplied to the generator at a flow rate of 6–17 L/min using a five-stage desiccant operated by an MFC, which was connected to the inlet of the TDA-4B Lite aerosol generator. The oil generator outlet was attached to a metal tube connected to the mixing chamber, where oil was generated and mixed with particle-free air. The amount of oil generated was controlled by changing the flow rate using the MFC.

#### 2.2.3. Experimental Design

Three of each type of low-cost PM sensors were used to evaluate the precision of mass concentration measurements given by each sensor. In total, 12 low-cost PM sensors, 1 medium-cost pDR-1500 sensor, and 1 reference instrument, GRIMM MiniWRAS, were used. All the low-cost PM sensors and the pDR-1500 recorded measurements with a 1 s frequency, and the reference instrument recorded at a 1 min frequency.

For OPC-N3, PM_1_ was changed to PM_4_ using the OPC-N3 software when switching from environmental to occupational concentrations and vice versa, per the manufacturer’s instructions. SPS30 sensors were connected to the computer through a USB cable and were operated using Sensirion sensor viewer software, version 2.91 on a Windows computer (Sensirion, Stäfa, Switzerland). AirBeam2 was operated using a Galaxy J3 2016 (Samsung, Suwon-si, Korea) phone with the AirCasting app. GRIMM was operated using the MiniWRAS version 10.0 software (Grimm Aerosol Technik, Ainring, Germany). The PMS A003 sensors, inside GeoAir1 device, started sampling as soon as the device was connected to the power supply.

For the pDR-1500, a 37 mm glass microfiber filter (934-AH, Whatman, Marlborough, MA, USA) was weighed at the beginning and end of each experiment using an XPR Micro-Analytical Balance (Mettler-Toledo, LLC, Columbus, OH, USA). The filter was then used to correct PM_2.5_ (environmental settings) and PM_4_ (occupational settings) mass concentrations for the reference instrument. Then, the cyclones were selected based on the type of experiment. The blue cyclone, with a cut-point diameter of 2.5 µm, was chosen for environmental concentrations, and the red cyclone, with a cut-point diameter of 4 µm, was chosen for occupational concentrations. The cut-point of 2.5 µm was chosen to measure PM_2.5_ mass concentrations representing fine PM, such as soot, oil, diesel fume, and wood, present in the ambient air at low concentrations [18]. Similarly, a cut-point of 4 µm was selected to measure PM_4_ mass concentrations, which represent coarse PM found at high concentrations in industrial workplaces, landfills, and agriculture and construction sites [31]. During each experiment, the flow rate (L/min) of the pDR-1500 was noted. Flow rates of 1.52 L/min and 2.65 L/min were used for the desired cut-point diameters of 2.5 µm and 4.0 µm, respectively. Data were downloaded to the computer using the pDR-1500 software for analysis at the end of the experiment.

After all the sensors were turned on, they were operated in particle-free air for the first 5 min. The reference for clean air (<0.1 μg/m^3^) was taken from the pDR-1500. For each aerosol source, two experiments were performed. For environmental settings, eight concentrations were generated (0, 10, 15, 20, 25, 30, 35, and 40 μg/m^3^) based on the EPA [10] standard for PM_2.5_ (ambient air), which is 35 μg/m^3^ in a 24 h period. Similarly, for occupational settings, eight concentrations were generated (0, 50, 100, 250, 500, 1000, 1500, and 2000 μg/m^3^) based on the OSHA standards for respirable dust particulates otherwise not regulated, which is 5000 μg/m^3^ [11]. The concentrations were lower than OSHA’s maximum standards but were within the limits (one-tenth of the standards) at which industrial hygienists would take action to control pollutants [24].

### 2.3. Data Analysis

#### 2.3.1. Accuracy and Bias

Data collected from the sensors were averaged over 1 min and time paired with the reference instrument using Matlab R2019B. Scatterplots for 5 min average mass concentrations of PM_1_, PM_2.5_, PM_4_, and PM_10_ were plotted against the reference instrument. The correction factor for the reference instrument was obtained by dividing the gravimetrically measured mass concentration (PM_2.5_ and PM_4_) by the mean of the reference instrument mass concentrations over the same period. After this point, we used PM_2.5_mc and PM_4_mc to denote the miniWRAS-corrected concentrations for environmental and occupational settings and PM_2.5_ and PM_4_ to denote the uncorrected concentrations for environmental and occupational settings, respectively. Finally, the performance of the low-cost sensors and pDR-1500 was examined for accuracy, bias, and precision under various conditions. The slope, intercept, and Pearson coefficient (*r*) were calculated using a linear regression model to measure the accuracy of the low-cost sensors compared to the reference instrument.

The bias, indicating how well the low-cost and pDR-1500 sensors agreed with the reference instrument, was calculated for each minute using the following equation [11]:(1)$$B=\frac{1}{n}\sum \frac{y_{i}-x_{i}}{x_{i}}$$
where *y* is the estimated mass concentration of the low-cost sensors and pDR-1500, *x* is the estimated mass concentrations of the reference instrument, and *n* is the number of data pairs.

The coefficient of variation (*CV*) was used to measure the precision of the low-cost sensors. CV was used to capture the inter-sensor variability, where mass concentrations were calculated for each minute using the following equation [11]:*CV* = *σ*/*µ*(2)
where *σ* is the standard deviation and *μ* is the mean of the mass concentrations reported. *CV* percentage was used to determine the precision of the measurements reported by the low-cost sensors of the same brand name. According to Sousan et al. [11], *CV* values less than 10% are considered acceptable. In this study, *CV* percentage was not calculated for pDR-1500 because only one device for this brand was used in the experiment.

All these parameters were then compared to EPA and NIOSH acceptance criteria. An ideal linear regression requires a slope of 1, intercept of 0, and *r* of 1 [11]. According to EPA and NIOSH, a slope of 1.0 ± 0.1, an intercept of 0 ± 5 µg/m^3^ (EPA only), *r* ≥ 0.97, and bias percentage of ±10% (NIOSH only) are the current acceptable standards.

#### 2.3.2. Significance Test

A *t*-test was used to determine whether there was any significant difference in the slope values between environmental and occupational settings (*α* = 0.05). The slope indicated overestimation or underestimation of the mass concentrations reported by the low-cost sensors compared to the reference instrument. In this study, slope values were compared between environmental and occupational settings to determine if low-cost sensors needed to be calibrated differently for environmental and occupational settings.

#### 2.3.3. Particle Size Distribution

For the particle size distribution, the reference instrument reported the particle counts for the first 10 bins in electrical mobility equivalent diameter. This diameter was then converted into the geometric equivalent diameter using a shape factor of 1 for salt, 1.5 for ARD, and 1 for PAO-4 oil [11,36]. PAO-4 oil is a pure liquid oil that, once aerosolized, generates oil particles. In this study, those liquid aerosol particles were spherical; therefore, we assumed the shape factor for PAO-4 oil to be 1 [36]. The remaining bins reported particle counts in optical diameter, which were kept unchanged and were assumed to be equal to the volumetric diameter [24]. Similarly, for the OPC-N3, SPS30, and PMS A003, we assumed that the optical diameters were equivalent to the volumetric diameters [24]. AirBeam2 and pDR-1500 were excluded from this comparison because they did not report the particle counts.

## 3. Results

### 3.1. Sensor Response

The *x*-axis error bars in the scatterplots presented in this section depict the standard deviations of mass concentrations reported by the GRIMM MiniWRAS. The *y*-axis depicts the standard deviations of mass concentrations reported by the low-cost sensors. We highlight what is important in the figures to reduce redundancy between the graphs and text.

For salt aerosol, the mass concentrations representing environmental settings of PM_1_, PM_2.5_mc, and PM_10_ (Figure 2) and occupational settings of PM_1_, PM_2.5_, PM_4_mc, and PM_10_ (Figure 3) for the low-cost sensors and pDR were compared to the MiniWRAS measures. The SPS30 showed a strong correlation with the reference instrument, with PM_1_, PM_2.5_, PM_4_mc, and PM_10_ measurements close to the 1:1 line for both environmental and occupational settings, but underestimated for PM_2.5_mc for environmental settings. For the ARD aerosol, the OPC-N3 and pDR-1500 followed the same pattern for environmental settings (Figure 4) and occupational settings (Figure 5). The OPC-N3 and pDR-1500 PM_2.5_mc overestimated mass for environmental settings, and PM_4_mc mass concentrations were on the 1:1 line for occupational settings. For PAO-4 oil aerosol, the OPC-N3 and SPS30 were noticeably close to the 1:1 line for specific particle sizes and settings. The OPC-N3 PM_1_, SPS30 PM_1_, and SPS30 PM_2.5_mc in environmental settings (Figure 6), and the OPC-N3 PM_4_mc and SPS30 PM1 in occupational settings (Figure 7) were close to the 1:1 line in their respective figures.

### 3.2. Sensor Accuracy and Bias: Environmental Evaluations

The mass concentrations of PM_1_, PM_2.5_, PM_4_, and PM_10_ reported by the sensors were compared to the reference instrument for salt, ARD, and PAO-4 oil for environmental and occupational concentrations. The results of the environmental evaluation are presented in Table 2. Similar to the previous section, we highlight what is important in the tables to reduce redundancy.

#### 3.2.1. OPC-N3

For all the aerosol sources, the mass concentrations of PM_1_, PM_2.5_, and PM_10_ reported by OPC-N3 were highly linear with the reference instrument (*r* = 0.99). These *r* values indicate that OPC-N3 distinctly met the EPA and NIOSH criteria. However, there was a significant variation in slope values for all aerosol sources and PM metrics. Noticeably, slope values met the performance criteria for ARD PM1, PAO-4 oil PM_1_, and PM_10_. For ARD, intercept values for all the aerosol sources and PM metrics except for PM_10_ (−7.04) were within the range of 0 ± 5 µg/m^3^, thus meeting the EPA criteria. Bias percentage was within the range of ±10% for ARD PM_1_ and PAO-4 oil PM_1_ and PM_10_.

#### 3.2.2. SPS30

For all the aerosol sources, mass concentrations of PM_1_, PM_2.5_, and PM_10_ reported by SPS30 were highly linear with the reference instrument, with *r* values greater than 0.98 except for ARD PM_10_ (*r* = 0.94). These *r* values indicate that SPS30 distinctly met the EPA and NIOSH criteria (except for ARD PM_10_). The slope values for salt PM_1_ and PM_10_ and PAO-4 oil PM_1_ and PM_2.5_mc were 0.92, 1.09, 1.1, and 0.90, respectively, which met EPA and NIOSH performance criteria. However, there was a significant variation in the slope for other aerosols and sizes. Intercept values for all the aerosol sources and PM metrics were within the range of 0 ± 5 µg/m^3^, which met the EPA criteria. Bias percentage was within the range of ±10% for salt PM_1_, salt PM_2.5_mc, and PAO-4 oil PM_1_.

#### 3.2.3. AirBeam2

For all sizes of salt and for PAO-4 oil PM_2.5_ and PM_10_, results were highly linear with the reference instrument (*r* ≥ 0.97). The overall calculated slopes and bias values for AirBeam2 for all aerosol sources and PM metrics did not meet the performance criteria. Intercept values for all the aerosol sources and PM metrics were within the range of 0 ± 5 µg/m^3^, which met the EPA criteria.

#### 3.2.4. PMS A003

For salt PM_1_ and ARD PM_2.5_mc, the results were highly linear with the reference instrument (*r* ≥ 0.97). Similar to the AirBeam2, the slopes and bias values for PMS A003 for all aerosol sources and PM metrics failed to meet the performance criteria. Intercept values for all the aerosol sources and PM metrics were within the range of 0 ± 5 µg/m^3^, which met the EPA criteria, with the exception of PAO-4 oil PM_10_ (9.76 µg/m^3^).

#### 3.2.5. pDR-1500

The pDR-1500 results were highly linear with the reference instrument, with *r* values greater than 0.98 for salt, ARD, and PAO-4 oil for PM_2.5_mc, which distinctly fulfills the EPA and NIOSH criteria. The slope value was closer to unity for salt (slope = 0.98), thus meeting the EPA and NIOSH performance criteria. Intercept values for all the aerosol sources were within the range of 0 ± 5 µg/m^3^, which also met the EPA criteria. The bias value for PM_2.5_mc met the EPA criteria.

### 3.3. Sensor Accuracy and Bias: Occupational Evaluations

The results of the occupational evaluations are shown in Table 3. Similar to the previous section, we highlight what is important in the tables to reduce redundancy.

#### 3.3.1. OPC-N3

For all aerosol sources, the results were highly linear with the reference instrument (*r* > 0.97) except for PM_1_. These *r* values show that OPC-N3 distinctly met the EPA and NIOSH criteria. Slope values met the performance criteria for PM_1_, PM_2.5_, PM_4_mc, and PM_10_, for salt, ARD, and PAO-4 oil, except for ARD PM_1_. Moreover, intercept values for PM_10_ for salt and PM_2.5_ and PM_4_mc for PAO-4 oil were within the range of 0 ± 5 µg/m^3^, which met the EPA criteria. Bias percentages were within the range of ±10% for ARD PM_2.5_, ARD PM_4_mc and PAO-4 Oil PM_4_mc.

#### 3.3.2. SPS30

The results for all PM metrics for all the aerosol sources were highly linear with the reference instrument (*r* = 0.99), indicating that SPS30 distinctly met the EPA and NIOSH criteria. However, there was a significant variation in the slope for all aerosol sources and PM metrics. For slope, only ARD PM_4_mc and PAO-4 oil PM_1_ met the EPA and NIOSH criteria. For intercept, only values for PM_1_ measurements for ARD and PAO-4 oil were within the range of 0 ± 5 µg/m^3^, which met the EPA criteria. Bias percentages were within the range of ±10% only for PM_1_ salt and PAO-4 oil.

#### 3.3.3. AirBeam2

For all the aerosol sources, the results were highly linear with the reference instrument (*r* = 0.99). For slope, only the PM_10_ measurement for salt met the EPA and NIOSH criteria. Intercept values for all the aerosol sources and PM metrics (excluding salt PM_10_ and PAO-4 oil PM_10_) were within the range of 0 ± 5 µg/m^3^, which met the EPA criteria. Bias percentages for all the aerosol sources and PM metrics did not meet the EPA and NIOSH criteria.

#### 3.3.4. PMS A003

For all the aerosol sources except PAO-4 oil (*r* = 0.96), the results for PM_1_, PM_2.5_, and PM_10_ were highly linear with the reference instrument, with *r* values greater than 0.98. Based on EPA and NIOSH criteria, the overall slopes for PMS A003 for all aerosol sources and PM metrics failed to meet the accuracy criteria. Intercept values for all the other aerosol sources and PM metrics were within the range of 0 ± 5 µg/m^3^, except for PM_2.5_ and PM_10_ for salt and PM_1_, PM_2.5_, and PM_10_ for PAO-4 oil. Bias percentages for all the aerosol sources and PM metrics did not meet the EPA and NIOSH criteria.

#### 3.3.5. pDR-1500

The pDR-1500 results were highly linear with the reference instrument, with *r* values greater than 0.99 for salt, ARD, and PAO-4 oil for PM_4_, which distinctly fulfilled the EPA and NIOSH criteria. For slope, only ARD met the performance criteria. For intercept, only the values for salt were within the range of 0 ± 5 µg/m^3^, which met the EPA criteria. Bias percentages for all the aerosol sources and PM metrics did not meet the EPA and NIOSH criteria.

### 3.4. Precision of Low-Cost Sensors

For environmental concentrations for all aerosol sources and PM metrics, the precision for OPC-N3 and AirBeam2 was low because the *CV* values did not meet the EPA criteria (*CV* < 10%). However, the precision for SPS30 for all sizes of salt was high because the *CV* values were less than 10%. For all PM metrics for ARD and PAO-4 oil, precision was greater than 10% (range: 10.77–20.36%), but *CV* values were only 10% higher than the EPA criteria. The precision for PMS A003 was high only for salt PM_2.5_ and PM_10_ with a *CV* of 8.66 and 4.66, respectively. For salt PM_1_; ARD PM_1_, PM_2.5_, and PM_10_; and PAO-4 oil PM_1_, PM_2.5_, and PM_10_, the *CV* was higher than 10%, failing to meet the EPA criteria for precision.

For occupational concentrations for all aerosol sources and PM metrics, precision for OPC-N3 was low because *CV* values were greater than 10%. However, precision for SPS30 was high, with *CV* values less than 10% for salt. For all PM metrics for ARD and PAO-4 oil, precision was greater than 10%, but percent *CV* values were 10% higher than the EPA criteria. For the AirBeam2, precision was high only for salt PM_2.5_ and PM_10_ and PAO-4 oil PM_2.5_ (*CV* < 10%). The precision for PMS A003 was high for salt PM_1_, PM_2.5_, and PM_10_, with *CV* values of 9.96%, 6.59%, and 8.16%, respectively. For ARD PM_1_, PM_2.5_, and PM_10_ and PAO-4 oil PM_1_, PM_2.5_, and PM_10_, *CV* values were higher than 10%, failing to meet the EPA criteria precision.

### 3.5. Significance Test

Table 4 shows a slope comparison between environmental and occupational settings for salt, ARD, and PAO-4 oil for PM_1_, PM_2.5_, and PM_10_ measurements. For OPC-N3, SPS30, and AirBeam2 for all the PM metrics and aerosol types, the *p* values were less than 0.05, indicating a significant difference between slope values for the environmental and occupational settings. The only exception was for salt PM_1_ for SPS30. For PMS A003, a significant difference was found between slope values (*p* < 0.05) among environmental and occupational settings for all PM metrics and aerosol types except for salt PM_1_ and ARD PM_1_.

### 3.6. Particle Size Distribution

For the particle size distribution, the low-cost sensors and the reference instrument number concentrations are shown in Appendix A for salt, ARD, and PAO-4 oil in environmental and occupational settings. For salt, the number concentrations measured by the reference instrument showed three peaks at 0.05 µm, 0.2 µm, and 0.3 µm. However, the number concentrations measured by OPC-N3 and PMS A003 showed unimodal peaks at 0.4 µm and 0.3 µm, respectively. For SPS30, bimodal distribution was observed, with peaks at 0.5 µm and 2.5 µm. The particle size distribution for environmental and occupational concentrations was found to be similar; the only noticeable difference was in the particle counts.

For ARD, the number concentrations measured by the reference instrument showed three peaks at 0.07 µm, 0.1 µm, and 0.4 µm. Similarly, bimodal distribution was observed for the number concentrations measured by OPC-N3 (peaks at 0.41 µm and 1.50 µm), SPS30 (peaks at 0.50 µm and 2.50 µm), and PMS A003 (peaks at 0.30 µm and 2.50 µm). The particle size distribution for environmental and occupational concentrations was found to be similar. The only difference noticed was the particle counts.

For PAO-4 oil, the number concentrations measured by the reference instrument showed four peaks at 0.05 µm, 0.19 µm, 0.40 µm, and 2.33 µm. Similarly, bimodal distribution was observed for the number concentrations measured by OPC-N3 (peaks at 0.41 µm and 1.50 µm), SPS30 (peaks at 0.50 µm and 2.50 µm), and PMS A003 (peaks at 0.30 µm and 2.50 µm).

The particle size distribution for environmental and occupational concentrations was found to be similar. The only difference noticed was in the particle counts.

## 4. Discussion

Low-cost PM sensor calibration is crucial for establishing reliable mass concentration estimates for different conditions and aerosol types. This study presents these estimates with accuracy and bias statistical analysis to assess the reliability of the data for practical use. In addition, this work highlights the necessity of calibrating these sensors based on their actual environment and posits that a universal calibration regression model cannot be used to estimate mass concentrations for both low and high mass concentrations. Previous studies have focused on using regression models for specific aerosol types at certain environmental or occupational conditions based on the applications in the literature. The current work extends this by presenting the difference between environments for different aerosol types and establishing the necessity of deriving a regression model for different concentration levels for the same aerosol type. These environments are not unique to environmental and occupational settings alone but also include indoor settings that could reach several hundred micrograms based on different indoor activities such as cooking, smoking, and various home improvement activities.

This study compared the mass concentrations (PM_1_, PM_2.5_, PM_4_, and PM_10_) reported by four low-cost sensors and one medium-cost sensor to those measured with the reference instrument GRIMM MiniWRAS for salt, ARD, and PAO-4 oil aerosols. The details are discussed below.

### 4.1. Response, Accuracy, and Bias of Sensors

#### 4.1.1. OPC-N3 in Environmental Settings

Similar to other studies, a high correlation was found between the mass concentrations reported by OPC-N3 and the reference instrument. In a laboratory evaluation conducted by the SCAQMD [37] with ARD, OPC-N3 showed a high correlation (*R*^2^ > 0.99) between all PM metrics compared with the reference instrument (GRIMM). Another study conducted by Li et al. [18] also observed a high correlation for sea salt (*R*^2^ > 0.92) and ARD (*R*^2^ > 0.95) for PM_2.5_ measurements compared to GRIMM.

Similarly, Sousan et al. [24] observed high linearity (*r* ≥ 0.97) between PM measurements reported by OPC-N2 and gravimetrically corrected SMPS and APS PM measurements for salt, welding fume, and ARD, with variations in slope values. For salt, the authors observed lower slope values for PM_1_ at 0.2 and PM_10_ at 0.5, indicating underestimation, which contrasted with our findings. This discrepancy in slope values might have occurred due to the different reference instruments used in the studies.

Moreover, for all aerosol sources in the present study, the intercept values for most PM metrics were low (closer to 0), meeting the EPA recommendations. The bias percentages were significantly higher for salt PM_1_ (failing to meet EPA and NIOSH criteria), indicating poor agreement between OPC-N3 and the reference instrument. In contrast, bias percentages were lower for ARD PM_1_ and PAO-4 oil PM_1_ (meeting EPA and NIOSH criteria) for all PM metrics and aerosol types, indicating good agreement between OPC-N3 and the reference instrument. It is worth noting that the OPC-N3 results were on the 1:1 line for all the PM metrics for PAO-4 oil but were overestimated for salt and underestimated for ARD, which might be due to differences in the properties of these particles (e.g., shape, density, and refractive index). Overall, these results suggest that OPC-N3 might be appropriate for measuring PAO-4 oil and ARD but not salt.

#### 4.1.2. OPC-N3 in Occupational Settings

As explained above, the overestimation of mass concentrations observed for occupational settings may have been due to the large calibration factor used by the manufacturer to calibrate the sensors [18], and underestimation might have resulted from low particle detection efficiency for smaller particles. The intercept values for salt (except PM_10_) and ARD were significantly high for all the PM metrics and thus did not meet the EPA recommendation. This is similar to Sousan et al. [24] findings for PM_2.5_ and PM_10_. In contrast, the intercept values for PAO-4 oil were low for PM_2.5_ and PM_4_, meeting the EPA recommendations, but were high for PM_1_ and PM_10_.

Despite the high percentage bias for all aerosols, ARD and PAO-4 oil exhibited lower values than salt. To our knowledge, we are the first to evaluate OPC-N3 in occupational settings. However, Sousan et al. [24] evaluated its predecessor, OPC-N2, for occupational concentrations and found a high bias (−19% to −92%) between PM measurements reported by OPC-N2 and gravimetrically corrected SMPS and APS PM measurements for salt, welding fume, and ARD.

In conclusion, based on the results for environmental and occupational settings, the OPC-N3 might be suitable for measuring ARD and PAO-4 oil for PM_1_ and PM_2.5_ concentrations in environmental settings but not for salt. Overall, the large deviation in slope, intercept, and percent bias indicated that OPC-N3 has low accuracy and might not be appropriate for mass concentration measurements in occupational settings. Therefore, it is apparent that OPC-N3 sensors need specific calibration for different settings.

#### 4.1.3. SPS30 in Environmental Settings

Tryner et al. [25] found good agreement (*R*^2^ ≥ 0.98) among SPS30 sensors and TEOM for PM_2.5_ measurements of ammonium sulfate (concentration < 1025 µg/m^3^) and ARD (concentration <540 µg/m^3^), which is similar to our findings. For salt, slope values were closer to unity for most PM metrics, meeting EPA and NIOSH performance criteria. For ARD, the slope value for the PM_1_ and PM_2.5_ measurement indicated overestimation, while underestimation was found for PM_10_ mass concentrations. Similarly, for PAO-4 oil, the slope value was closer to unity for the PM_1_ and PM_2.5mc_ measurement, meeting EPA and NIOSH performance criteria. However, underestimation was observed for PM_2.5_ and PM_10_.

SPS30 is factory-calibrated with atomized potassium chloride, similar to the salt we used in this study, which may explain the high correlation of SPS30 with salt [25]. In addition, the underestimation of PM_2.5_ mass concentration by SPS30 for ARD was different with the result found by Tryner et al. [25], who compared SPS30 with TEOM for concentrations up to 1000 µg/m^3^. The authors also reported that SPS30 was highly accurate in sorting all the particulate mass to the PM_1_ bin relative to the APS, which might explain the slope value for the PAO-4 oil PM_1_ measurement being closer to unity.

Similarly, Kuula et al. [20] also showed a high detection range (<0.9 µm) for the first bin (0.3–1.0 µm), which suggests that SPS30 correctly classifies particles less than 1 µm into the first bin. The authors also found that the accuracy for mass concentration measurements decreased for larger particle sizes (PM_2.5_ and PM_10_), which might explain the underestimation of PM_2.5_ and PM_10_ for ARD and PAO-4 oil in this study.

Overall, these observations indicate that SPS30 might perform better for salt than for ARD or PAO-4 oil in environmental settings. The findings also suggest that SPS30 should be calibrated for ARD and PAO-4 oil to improve its performance, especially in environmental settings.

#### 4.1.4. SPS30 in Occupational Settings

The occupational results indicated that SPS30 has moderate accuracy for PM_1_ measurements for all aerosol types, but it may not be appropriate for PM_2.5_, PM_4_, or PM_10_ measurements. Moreover, SPS30 would be more suitable for measuring salt aerosol than ARD or PAO-4 oil due to its factory calibration.

In conclusion, the results for the environmental and occupational settings show that SPS30 might be suitable for all aerosol types in environmental settings, but especially for salt. Overall, the large deviation in slope, intercept, and percent bias indicated that SPS30 has low accuracy and might not be appropriate for mass concentration measurements in occupational settings. Therefore, it is apparent that SPS30 sensors require specific calibration for different settings.

#### 4.1.5. AirBeam2 in Environmental Settings

Our finding were similar to those of the manufacturer [27], which exhibited a moderate correlation (*R*^2^ = 0.88 for PM_1_ and *R*^2^ = 0.89 PM_2.5_) between AirBeam2 and TSI DustTrak DRX Aerosol Monitor 8533. Similarly, SCAQMD [37] found a moderate correlation (*R*^2^ = 0.72 for PM_1_ and *R*^2^ = 0.64 for PM_2.5_) between AirBeam2 and GRIMM; however, underestimation was observed for all the PM metrics, which is consistent with our findings. The researchers also found that AirBeam2 does not correlate well (*R*^2^~0) with GRIMM for PM_10_ measurements, indicating that AirBeam2 may not be appropriate for obtaining mass concentration measurements for large particles [27]. There was also a significant variation in slope values.

The reason for the underestimation of all PM metrics, except ARD PM_10_ might be due to the calibration equation used by the manufacturer to calculate mass concentrations. The manufacturer obtained this equation by comparing PurpleAir-I (PMS 1003 sensors) and PurpleAir-II (PMS 5003 sensors) with GRIMM measurements instead of using PMS 7003, which is what is used in the AirBeam2 device [27].

However, one important difference worth noting in this study is that slope values were lower than unity, especially for ARD and PAO-4 oil, which were extremely low (closer to 0). These results showed that to some extent, AirBeam2 might perform better for salt in environmental settings than for ARD or PAO-4 oil.

#### 4.1.6. AirBeam2 in Occupational Settings

AirBeam2 is calibrated only for low concentrations, so we observed considerable differences in the mass concentration measured by AirBeam2 in occupational settings compared to environmental settings, which explains the underestimation of mass concentrations [27]. Overall, these observations indicated that AirBeam2 might perform better for salt than ARD and PAO-4 oil in occupational settings.

In conclusion, the results for the environmental and occupational settings show that AirBeam2 might be suitable for salt in environmental settings to some extent. Overall, the large deviation in slope and intercept values indicated that AirBeam2 has low accuracy and might not be appropriate for mass concentration measurements in the occupational setting. Therefore, it is apparent that AirBeam2 sensors need specific calibration for different settings. Moreover, AirBeam2 may need a better calibration factor than the one used by the manufacturer to calibrate these sensors.

#### 4.1.7. PMS A003 in Environmental Settings

Our accuracy results were consistent with those of Levy Zamora et al. [3], who found a high correlation for ARD (*R*^2^ = 0.96) and salt (*R*^2^ = 0.99) between PMS A003, APS, and SMPS for PM_2.5_ measurements but low accuracy overall. They also found that PMS A003 did not sort the particles correctly into their respective bins compared to APS, which may explain the large underestimation of mass concentration found in this study.

Another reason for underestimation may be that this study used a different aerosol type than the one used by the manufacturer to calibrate the sensors. However, one important difference worth noting was that slope values for salt were on the higher side (closer to unity) than the slope values for ARD and PAO-4 oil, which were extremely low (closer to 0). These results show that PMS A003 might perform better for salt than ARD or PAO-4 oil in environmental settings.

A high percent bias for all aerosols indicates overestimation of mass concentrations compared to the reference instrument. PMS A003 exhibited lower percent bias values for salt compared to ARD and PAO-4 oil. These observations indicate that PMS A003 might perform better for salt than ARD or PAO-4 oil in environmental settings.

#### 4.1.8. PMS A003 in Occupational Settings

Similar to environmental settings, underestimation of particles was also observed in occupational settings. The different responses observed between aerosol types may be due to differences in the properties of the aerosols, such as shape, refractive index, and density. The density of salt was 2200 kg/m^3^, ARD was 2650 kg/m^3^, and PAO-4 oil was 800 kg/m^3^. Overall, the deviation in slope and percent bias found in this study indicate that PMS A003 might not be appropriate for mass concentration measurements in occupational settings.

In conclusion, the results for environmental and occupational settings show that PMS A003 might be suitable for salt, but only in environmental settings. However, correction factors may make PMS A003 suitable for ARD and PAO-4 oil as well. Overall, the large differences observed in the slope values (except for salt and ARD PM_1_ measurements) indicate that PMS A003 has low accuracy and might not be appropriate for mass concentration measurements in occupational settings. Therefore, PMS A003 sensors require specific calibration for different settings.

#### 4.1.9. pDR-1500 in Environmental and Occupational Settings

The manufacturer calibrated pDR-1500 for dust particles, as indicated by the low bias agreement for ARD compared to the other two aerosols. Sousan et al. [11] also found high correlation (*r* = 0.9) for pDR-1500 compared to the reference instruments for salt and ARD in occupational settings. However, Sousan et al. [11] used a cyclone with a cut-off diameter of 10 µm compared to 4 µm used in this study. In conclusion, the results for environmental and occupational settings show that pDR-1500 might be suitable for salt, PAO-4 oil, and ARD in both environmental and occupational settings.

### 4.2. Precision of Low-Cost Sensors

For the OPC-N3, the results of this study were similar to those found by Badura et al. [38], who showed low precision (*CV* = 20%) for OPC-N2. Similarly, Sousan et al. [24] found low precision (*CV* = 16%) for ARD but high precision for salt (*CV* < 6%). These findings indicate that not all OPC-N3 devices can be treated equally, and each OPC-N3 may require specific calibration for the precise mass concentration measurements. To our knowledge, no other studies have described the precision of OPC-N3 so far.

SPS30 exhibited higher precision for all PM metrics and aerosol types compared to the other sensors used in this study. The reason for this might be the use of potassium chloride for the calibration of SPS30 sensors. Similarly, Tryner et al. [25] compared eight different SPS30 sensors using five different aerosols: ammonium sulfate, ARD, National Institute of Standards and Technology urban PM, oil mist, and PSL particles. The authors also found high precision (relative standard deviation < 10%) among SPS30 sensors for PM_2.5_ mass concentration measurements.

SCAQMD [37] found high precision among the AirBeam2 sensors, which contrasted with our findings. For PMS A003, Levy Zamora et al. [3] found high precision (*CV* = 12%) for the salt PM_2.5_ measurement, indicating high reproducibility among sensors, which contrasted with our findings. We found that the precision of PMS A003 for salt for all PM metrics was relatively low compared to PAO-4 oil and ARD, which is consistent with Levy Zamora et al. [3] findings to some extent. The low precision of AirBeam2 and PMSA003 indicates that these sensors are not calibrated equally by the manufacturer.

### 4.3. Significance Test

The results indicated that the sensors should be calibrated differently for different particle sizes when used for low and high concentrations. The three anomalies presented for SPS30 and PMS A003 for PM_1_ could be due to the slope values exhibiting similar behavior at low concentrations between environmental and occupational settings.

### 4.4. Particle Size Distribution

The sensors tested in this study had a lower total particle number concentration for salt, ARD, and PAO-4 oil for both environmental and occupational concentrations compared to the reference device, MiniWRAS. The exception was for PMS A003, which detected a higher number of 1 µm particles than the reference device. MiniWRAS has more bins (41) than the low-cost sensors (OPC-N3 = 6 bins, SPS30 = 5 bins, and PMS A003 = 6 bins), which explains its superior ability to capture the size distribution compared to the low-cost sensors. MiniWRAS can detect particles as small as 0.01 µm in diameter, whereas the low-cost sensors cannot detect particles smaller than 0.30 µm, explaining the underestimation observed.

## 5. Conclusions

This study evaluated the performance of four low-cost sensors (OPC-N3, SPS30, AirBeam2, and PMS A003) and one medium-cost instrument (pDR-1500) compared to the MiniWRAS reference instrument for measuring environmental and occupational mass concentrations (PM_1_, PM_2.5_, PM_4_, and PM_10_) of various aerosol types (salt, ARD, and PAO-4 oil) in a laboratory setting. Among all the low-cost sensors, SPS30 and OPC-N3 demonstrated the best performance, with high correlation and the lowest bias values, for all aerosol types and PM metrics in environmental and occupational settings. SPS30 exhibited high accuracy, particularly for salt aerosol PM_1_ and PM_2.5_ in environmental and occupational settings. However, the linear performance of SPS30 for ARD and PAO-4 suggests that aerosol-specific calibration may be needed to improve its measurement accuracy in environmental settings. OPC-N3 exhibited high accuracy for PAO-4 oil (environmental settings only) and ARD (PM_2.5_ in environmental settings only). In contrast, AirBeam2 and PMS A003 exhibited low accuracy for all aerosol types and PM metrics in both settings.

Regarding intra-instrument precision, SPS30 and OPC-N3 sensors were more precise than AirBeam2 and PMS A003 for all aerosol types and PM metrics. Overall, the findings showed that low-cost sensors performed better and had higher accuracy compared to the reference instrument for environmental concentrations (up to 40 µg/m^3^) because these sensors are calibrated for lower concentrations. However, the *t*-test indicated a significant difference in the slope values between environmental and occupational settings. This suggests that low-cost sensors must be calibrated differently for occupational concentrations (up to 2000 µg/m^3^) to improve measurement accuracy.

This study was conducted in a laboratory setting with controlled temperature and relative humidity. The results, therefore, may not be applicable to field applications with ambient settings where these parameters are not controlled. Future research is therefore recommended for field evaluation of low-cost sensors. These sensors should be evaluated under the target conditions, and an appropriate correction factor should be developed prior to the field deployment of these sensors. This work can be used as a reference for air quality specialists in the field, which would help guide them to choose the best sensor for different applications and environments. However, site calibration and evaluation are still recommended because this work was performed under controlled humidity and temperature settings and specific aerosol generation. Different aerosol types might coexist in the same environment in real-world applications, and changes in outdoor humidity may affect sensor response, which was not addressed in this work.

## Figures and Tables

**Figure 1 sensors-21-04146-f001:**
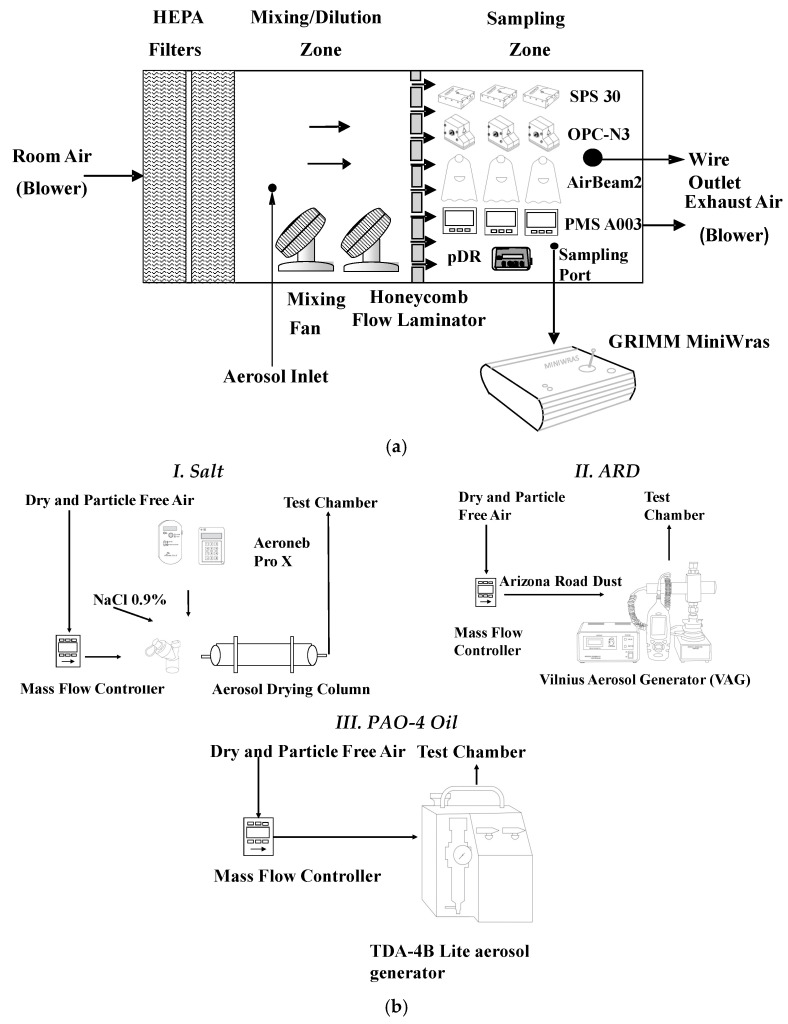
(**a**) Experimental setup and (**b**) aerosol generation systems used in the evaluation of the low-cost sensors. HEPA: high efficiency particulate air; NaCl: sodium chloride; ARD: Arizona road dust; VAG: Vilnius aerosol generator; PAO-4: poly-alpha-olefin-4.

**Figure 2 sensors-21-04146-f002:**
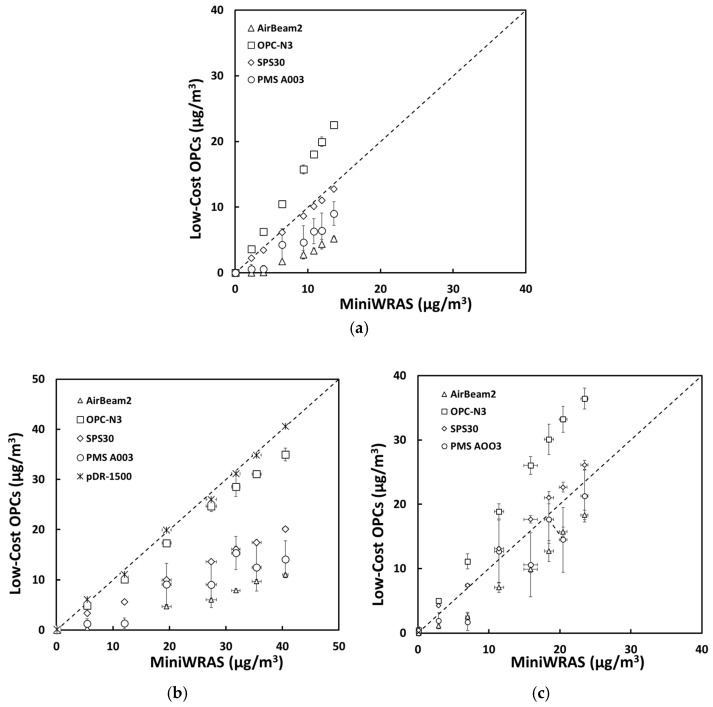
The mass concentrations (µg/m^3^) of (**a**) PM_1_, (**b**) PM_2.5_mc, and (**c**) PM_10_ reported by the sensors plotted against the reference device for the salt environmental experiment.

**Figure 3 sensors-21-04146-f003:**
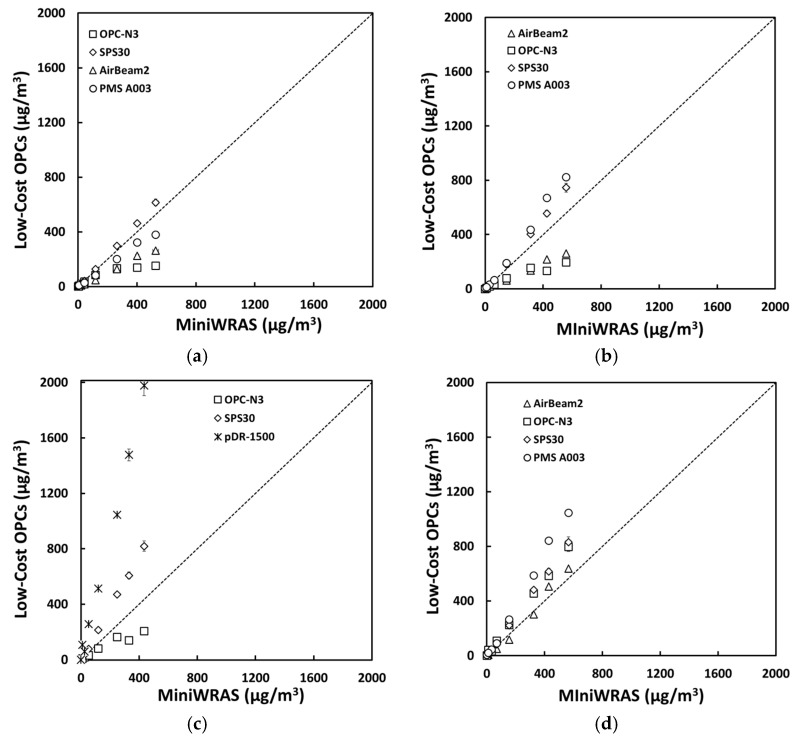
The mass concentrations (µg/m^3^) of (**a**) PM_1_, (**b**) PM_2.5_, (**c**) PM_4_mc, and (**d**) PM_10_ reported by the sensors plotted against the reference device for the salt occupational experiment.

**Figure 4 sensors-21-04146-f004:**
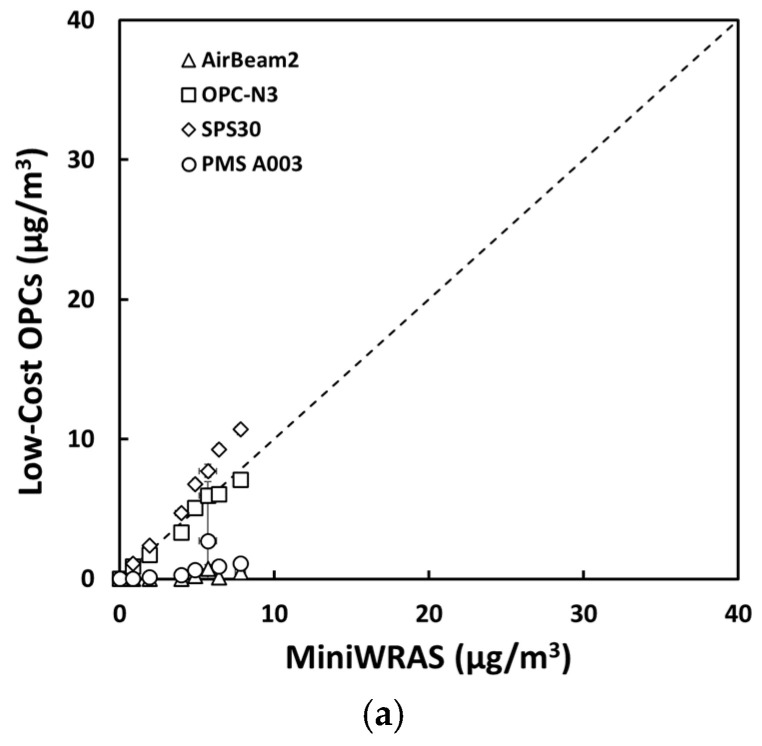
The mass concentrations (µg/m^3^) of (**a**) PM_1_, (**b**) PM_2.5_mc, and (**c**) PM_10_ reported by the sensors plotted against the reference device for the ARD environmental experiment.

**Figure 5 sensors-21-04146-f005:**
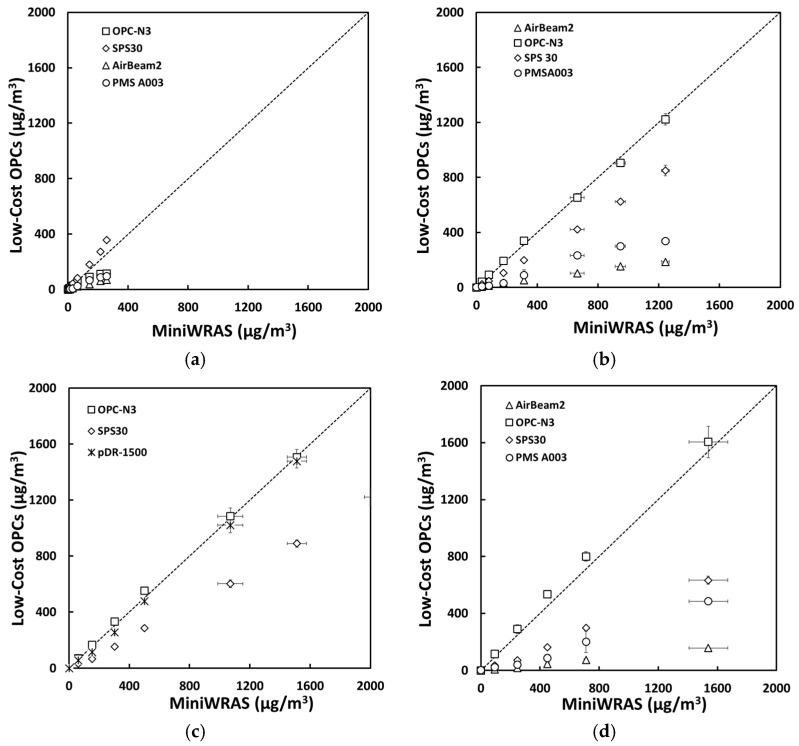
The mass concentrations (µg/m^3^) of (**a**) PM_1_, (**b**) PM_2.5_, (**c**) PM_4_mc, and (**d**) PM_10_ reported by the sensors plotted against the reference device for the ARD occupational experiment.

**Figure 6 sensors-21-04146-f006:**
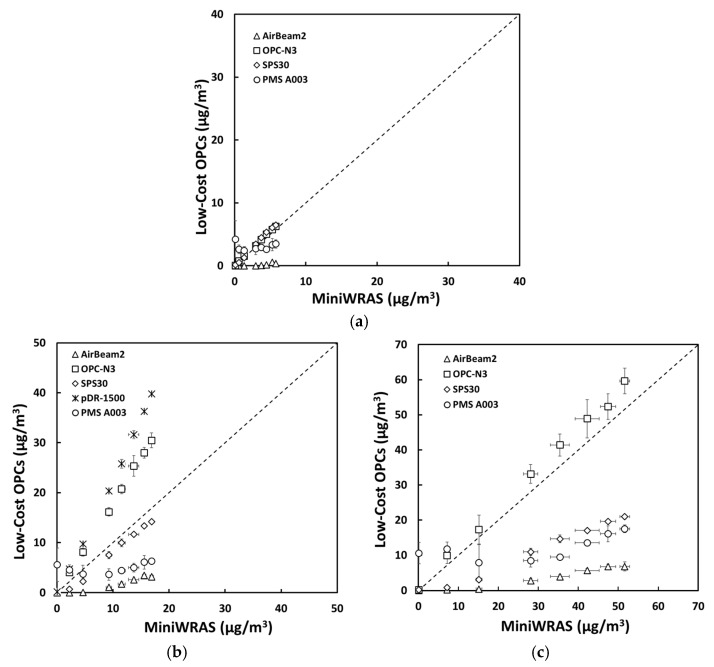
The mass concentrations (µg/m^3^) of (**a**) PM_1_, (**b**) PM_2.5_mc, and (**c**) PM_10_ reported by the sensors plotted against the reference device for the PAO-4 oil environmental experiment.

**Figure 7 sensors-21-04146-f007:**
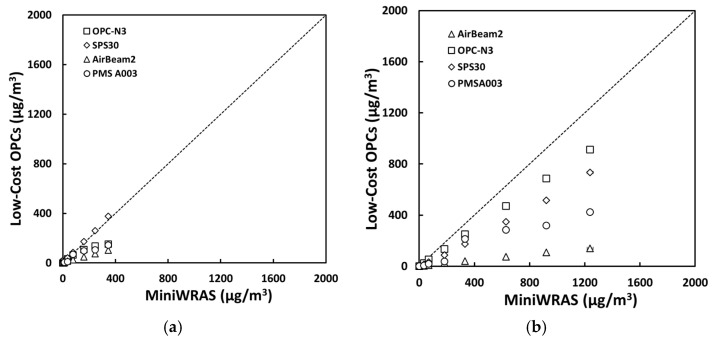
The mass concentrations (µg/m^3^) of (**a**) PM_1_, (**b**) PM_2.5_, (**c**) PM_4_mc, and (**d**) PM_10_ reported by the sensors plotted against the reference device for the PAO-4 oil occupational experiment.

**Table 1 sensors-21-04146-t001:** Specifications of the low-cost OPCs and pDR-1500.

Technical Data	OPC-N3	SPS30	AirBeam2	PMS A003	pDR-1500
Cost ($)	500	84	250	34	$7000
Size range (µm)	0.3–40.0	0.3–10.0	0.3–10.0	0.3–10.0	One size based on cyclone
Type (active or passive flow)	Active	Active	Active	Active	Active
Bin size (software bins, dimensionless)	24	5	6	6	-
Concentration range (µg/m^3^)	0–2000	0–1000	0–1000	0–1000	0.001–400,000
Mass concentration measurement	PM_1_, PM_2.5_, PM_4_, PM_10_	PM_1_, PM_2.5_, PM_4_, PM_10_	PM_1_, PM_2.5_, PM_10_	PM_1_, PM_2.5_, PM_10_	PM_1_, PM_2.5_, PM_4_, PM_10_
Number concentration	Yes	Yes	No	Yes	No
Sampling frequency	1 s	1 s	1 s	1 s	1 s
Internal fan	Yes	Yes	Yes	Yes	Pump
Additional air quality measurements	Temperature and relative humidity	NA	Temperature and relative humidity	Temperature and relative humidity	Temperature and relative humidity
Remote storage	No	No	Yes	No	No
Internal storage	Yes	No	No	No	Yes
Internal rechargeable battery	No	No	Yes	No	Yes
Dimensions: L × W × H (m)	0.075 × 0.006 × 0.060	0.0406 × 0.0406 × 0.0122	0.132 × 0.098 × 0.027	0.038 × 0.035 × 0.012	0.181 × 0.143 × 0.0484
Weight (g)	105	26	142	40	1200
WiFi connectivity	No	No	Yes	No	No
Visible output on sensor	No	No	No	No	Yes

OPC: optical particle counter; PM: particulate matter.

**Table 2 sensors-21-04146-t002:** Evaluation of mass concentrations reported by the sensors compared with the reference instrument for (A) PM_1_, (B) PM_2.5_mc, and (C) PM_10_ for environmental concentrations. EPA and NIOSH acceptance criteria: slope = 1.0 ± 0.1; intercept = 0 ± 5 µg/m^3^ (EPA only); *r* ≥ 0.97; and bias percentage = ±10% (NIOSH only).

Instruments	Data Pairs	Slope	Intercept	*r*	% Bias	% *CV*
**(A) PM_1_**						
**Salt**						
OPC-N3	8	1.67	0.12	0.99	64.81	74.06
SPS30	8	0.92	0.07	0.99	63.30	5.59
AirBeam2	8	0.41	0.79	0.97	−76.71	34.83
PMS A003	8	0.65	0.78	0.97	−61.10	11.38
**ARD**						
OPC-N3	8	0.21	0.007	0.99	−4.30	21.61
SPS30	8	1.40	−0.19	0.99	32.65	20.36
AirBeam2	8	0.06	−0.06	0.65	−96.44	91.16
PMS A003	8	0.21	−0.14	0.65	−86.18	105.77
**PAO-4 Oil**						
OPC-N3	8	1.09	−0.01	0.99	10.90	55.42
SPS30	8	1.15	−0.03	0.99	8.28	10.77
AirBeam2	8	0.07	−0.08	0.83	−96.80	105.92
PMS A003	8	0.08	3.47	0.34	35.50	97.51
**(B) PM_2.5_mc**						
**Salt**						
OPC-N3	8	0.87	0.05	0.99	−41.76	78.37
SPS30	8	0.49	0.31	0.99	−6.10	8.54
AirBeam2	8	0.29	−1.05	0.99	−79.48	34.83
PMS A003	8	0.38	−0.71	0.95	−70.00	8.66
pDR-1500	8	0.98	0.04	0.99	−0.24	-
**ARD**						
OPC-N3	8	2.91	−0.54	0.99	183.67	22.12
SPS30	8	1.43	0.67	0.99	52.84	18.00
AirBeam2	8	0.21	−0.11	0.90	−83.65	23.24
PMS A003	8	0.28	−0.46	0.97	−82.29	86.43
pDR-1500	8	2.14	2.27	0.98	150.09	-
**PAO-4 Oil**						
OPC-N3	8	1.82	−0.19	0.99	79.30	60.43
SPS30	8	0.90	−0.88	0.99	−28.23	11.39
AirBeam2	8	0.22	−0.52	0.97	−87.23	65.72
PMS A003	8	0.07	4.26	0.43	−32.00	70.37
pDR-1500	8	2.35	−0.71	0.99	123.60	-
**(C) PM_10_**						
**Salt**						
OPC-N3	8	1.58	0.47	0.99	−11.72	78.18
SPS30	8	1.09	0.48	0.99	16.57	8.93
AirBeam2	8	0.79	1.53	0.98	−39.63	22.65
PMS A003	8	0.89	1.04	0.94	−11.55	4.16
**ARD**						
OPC-N3	8	1.55	−7.04	0.99	39.51	27.56
SPS30	8	0.25	4.28	0.94	−64.92	15.09
AirBeam2	8	0.05	1.07	0.79	−93.02	30.27
PMS A003	8	0.11	0.81	0.94	−87.86	53.13
**PAO-4 Oil**						
OPC-N3	8	1.14	0.25	0.99	18.70	59.42
SPS30	8	0.43	−1.07	0.99	−66.32	12.41
AirBeam2	8	0.14	−0.68	0.97	−90.41	49.67
PMS A003	8	0.09	9.76	0.58	−46.62	47.55

*CV*: coefficient of variation.

**Table 3 sensors-21-04146-t003:** Evaluation of mass concentrations reported by the sensors compared with the reference instrument for (A) PM_1_, (B) PM_2.5_, (C) PM_4_mc, and (D) PM_10_ for occupational concentrations. EPA and NIOSH acceptance criteria: slope = 1.0 ± 0.1; intercept = 0 ± 5 µg/m^3^ (EPA only); *r* ≥ 0.97; and bias percentage = ±10% (NIOSH only).

Instruments	Data Pairs	Slope	Intercept	*r*	% Bias	% *CV*
**(A) PM_1_**						
**Salt**						
OPC-N3	8	0.29	20.22	0.88	−17.55	18.09
SPS30	8	0.17	−5.69	0.99	6.28	4.61
AirBeam2	8	0.52	−3.63	0.99	−55.40	11.09
PMS A003	8	0.74	2.07	0.99	−16.40	9.96
**ARD**						
OPC-N3	8	0.46	10.62	0.94	22.26	40.62
SPS30	8	1.32	−1.40	0.99	32.26	18.40
AirBeam2	8	0.27	0.37	0.99	−74.71	12.46
PMS A003	8	0.39	0.28	0.99	−63.41	51.75
**PAO-4 Oil**						
OPC-N3	8	0.46	12.95	0.94	16.47	37.59
SPS30	8	1.08	−0.11	0.99	9.29	19.63
AirBeam2	8	0.29	0.88	0.99	−71.61	17.81
PMS A003	8	0.42	7.22	0.96	−50.71	78.72
**(B) PM_2.5_**						
**Salt**						
OPC-N3	8	0.34	10.21	0.97	−52.89	20.64
SPS30	8	1.34	−11.49	0.99	13.74	6.77
AirBeam2	8	0.15	2.29	0.99	−53.67	6.48
PMS A003	8	0.29	−2.03	0.98	29.40	6.59
**ARD**						
OPC-N3	8	0.96	14.2	0.99	6.12	35.48
SPS30	8	0.68	−9.46	0.99	−36.58	18.70
AirBeam2	8	0.15	2.29	0.99	−83.73	51.49
PMS A003	8	0.29	−2.03	0.98	−74.26	47.74
**PAO-4 Oil**						
OPC-N3	8	0.74	2.72	0.99	−24.57	44.71
SPS30	8	0.59	−11.68	0.99	−48.41	19.68
AirBeam2	8	0.11	2.53	0.99	−87.32	10.69
PMS A003	8	0.35	12.47	0.96	−65.16	71.16
**(C) PM_4_mc**						
**Salt**						
OPC-N3	8	0.46	9.60	0.97	38.95	20.93
SPS30	8	1.89	−10.21	0.99	63.81	7.68
pDR-1500	8	4.47	−3.11	0.99	419.79	-
**ARD**						
OPC-N3	8	1.03	9.05	0.99	7.44	33.44
SPS30	8	0.61	−18.24	0.99	−45.16	19.23
pDR-1500	8	1.00	−28.27	0.99	−8.52	-
**PAO-4 Oil**						
OPC-N3	8	1.05	1.23	1.00	6.10	44.11
SPS30	8	0.83	−19.19	0.99	−29.14	19.70
pDR-1500	8	1.71	−32.83	0.99	52.08	-
**(D) PM_10_**						
**Salt**						
OPC-N3	8	1.38	4.79	0.99	63.34	117.93
SPS30	8	1.48	−9.76	0.99	28.24	6.85
AirBeam2	8	1.15	−23.22	0.99	−15.12	5.79
PMS A003	8	1.90	−16.55	0.99	63.54	8.16
**ARD**						
OPC-N3	8	1.07	13.18	0.99	12.75	28.56
SPS30	8	0.45	−24.11	0.99	−46.11	19.49
AirBeam2	8	0.10	−2.01	0.99	−85.84	50.14
PMS A003	8	0.27	−1.09	0.98	−63.85	40.53
**PAO-4 Oil**						
OPC-N3	8	0.84	6.84	0.99	−12.00	42.35
SPS30	8	0.59	−20.62	0.99	−49.83	19.09
AirBeam2	8	0.23	−8.62	0.99	−71.26	11.95
PMS A003	8	0.45	25.50	0.96	−32.12	61.56

**Table 4 sensors-21-04146-t004:** Results of the *t*-test to compare slopes for environmental and occupational settings.

	Salt			ARD			PAO-4 Oil		
	PM_1_	PM_2.5_	PM_10_	PM_1_	PM_2.5_	PM_10_	PM_1_	PM_2.5_	PM_10_
OPC-N3	<0.001	<0.001	<0.001	0.053	<0.001	<0.001	0.001	<0.001	<0.001
SPS30	<0.001	<0.001	<0.001	0.211	<0.001	<0.001	0.041	<0.001	<0.001
AirBeam2	0.024	<0.001	<0.001	<0.001	0.02	0.0095	<0.001	<0.001	<0.001
PMS A003	0.218	<0.001	<0.001	0.101	<0.001	<0.001	<0.001	<0.001	<0.001

Note: green indicates significant difference and red indicates no difference between environmental and occupational settings.

## Data Availability

The datasets generated from the current study are available from the corresponding author on reasonable request.

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
