# Peer review of "Laboratory Evaluation of Low-Cost Optical Particle Counters for Environmental and Occupational Exposures"

_sensors, 2021, doi:10.3390/s21124146_

Round 1

Reviewer 1 Report

The purpose of this paper is to provide new results for existing commercial PM sensors. Although the authors use some statistical methods, they seem to have written some simple instructions for the user. The paper is expressed redundantly and not concisely, and some details are not provided.

  1. The abstract may include how to calibrate sensors  for low performances one, which could be more intersting for the audience of sensor.
  2. The  introduction part should be improved by citing the more latetest works, for example
    Wang Y, Xu Z. Monitoring of PM2.5 Concentrations by Learning from Multi-Weather Sensors. Sensors. 2020; 20(21):6086. https://doi.org/10.3390/s20216086
  3. Please plots some illustrates for OPC-N3, AirBeam2, SPS30, PMS A003, pDR-1500 in Section 2.
  4. Please wirte some more information for GRIMM, and explain the full name of GRIMM and why the authors use it as reference sensor?
  5. Could you please provide the field photos for the experimental setup, which is more reliable for review.
  6. Why the data acquised by low cost sensor is not coherent with MiniWARS? 

Author Response

Please see attached response for all reviewers.

Reviewer 2 Report

The topic is interesting and is important for the scientific community and for the managers responsible to define parameters for monitoring indoor and outdoor air quality. 

GENERAL COMMENTS:

  • Extension of the manuscript: it would improve the scientific value if shortened.
  • If applicable, insert the highlights.
  • Review the keywords to increase the paper reach.
  • More detailed discussion is needed. In “Discussion Section” it is not clear the real contribution of the paper.
  • It is interesting to mention how the specialists can use this research to develop some actions to improve air quality monitoring.
  • It is important to write some examples or explain the main characteristics of environmental and occupational settings.
  • The authors can put some graphs in a supplementary information.
  • The authors can insert information about EPA and NIOSH performance criteria and recommendations in the table 3.
  • Section 4.1: The authors can show in two topics: 4.1.1. Environmental Settings and 4.1.2. Occupational Settings
  • It is important to explain the difference between PMxx and PMXXmc (XX = 1.0,2.5,4.0,10) if there is some difference in the nomenclature.

SPECIFIC COMMENTS:

Line 53: verify “PM.”

Line 120: In table 1, it is important to explain “Type” and put the dimensions in “Bin size”.

Line 167: The sensor “pDR-1500” was not in table 1. You can explain why you describe some characteristics of this sensor.

Line 462: The bias percentage ranged from -82.25% to 782.27% is too large. Did you repeat the experiments? How can you explain the huge differences?

Line 532: The bias percentage ranged from -68.74% to 456.45% is too large. Did you repeat the experiments? How can you explain the huge differences?

For all graphs: I suggest that you have to put the units in the axis. The number indicates the mass concentrations (mg/m3);

Check the unit in the text: mg/m3. There is some unit that is write with μg/m3 and ug/m3

Author Response

(The authors gave the same response as above.)

Reviewer 3 Report

The idea behind the work is interesting and I am sure researchers involved in air quality monitoring using low-cost sensors would benefit from such a study.

My major concern is about the design of the study, especially the comparison between different solutions. On one hand, the authors mention that they want to evaluate low-cost Optical Particle Counters, but during there study they have used AirBeam2; which is not just a sensor but an IoT device with multiple sensors (PMS 7003). To keep the study consistent, I would suggest to rephrase their problem statement. Either they should use more devices (like Purple Air, AirBox etc.) or they should strictly keep it to using low-cost sensors.
Also, the authors use PMS A003 that belongs to the Plantower family, AirBeam 2 has PMS7003, but many state-of-the-art low-cost sensing devices use PMS5003 (PurpleAir, AirBox). I would like to ask the authors to explain the reason behind selecting two particular models from the Plantower range.

Regarding the visualization, the scatter plots should be imporved.

Did the authors find any relationship between PM count and, Temperature and RH?

What was the inter-sensor variability?

Author Response

(The authors gave the same response as above.)

Round 2

Reviewer 1 Report

In my opinion, the manuscript is suitable for publication in Sensors​, after the authors have addressed comments and questions raised.

Author Response

Please see the document attached. 

Reviewer 3 Report

I would like to thank the authors for addressing all the concerns highlighted in the first review. Some minor comments:

  • Line 167- It should be PMS7003, not PMS7000
  • Line 154- It should be PMS7003. Please double check the sensor codes as there are several instances where they are not correct.

Author Response

Please see the document attached. 
